# Reliability, Validity, and Feasibility of the Frail-VIG Index

**DOI:** 10.3390/ijerph18105187

**Published:** 2021-05-13

**Authors:** Anna Torné, Emma Puigoriol, Edurne Zabaleta-del-Olmo, Juan-José Zamora-Sánchez, Sebastià Santaeugènia, Jordi Amblàs-Novellas

**Affiliations:** 1Central Catalonia Chronicity Research Group (C3RG), Centre for Health and Social Care Research (CESS), Faculty of Medicine, University of Vic-Central University of Catalonia (UVIC-UCC), 08500 Barcelona, Spain; atorneco@gmail.com (A.T.); sebastia.santaeugenia@gencat.cat (S.S.); 2Geriatric and Palliative Care Department, Hospital Universitari de la Santa Creu and Hospital Universitari de Vic, 08500 Barcelona, Spain; 3Clinical Epidemiology Unit, Consorci Hospitalari de Vic, 08500 Barcelona, Spain; epuigoriol@chv.cat; 4Tissue Repair and Regeneration Laboratory (TR2Lab), Faculty of Sciences and Technology, Faculty of Medicine, University of Vic-Central University of Catalonia, 08500 Barcelona, Spain; 5Fundació Institut Universitari per a la Recerca a L’atenció Primària de Salut Jordi Gol I Gurina (IDIAPJGol), 08500 Barcelona, Spain; ezabaleta@idiapjgol.org; 6Gerència Territorial de Barcelona, Institut Català de la Salut, 08500 Barcelona, Spain; juanjozamora72@gmail.com; 7Nursing Department, Faculty of Nursing, Universitat de Girona, 17005 Girona, Spain; 8Chronic Care Program, Ministry of Health, Generalitat de Catalunya, 08830 Catalonia, Spain

**Keywords:** feasibility, frailty, frailty index, psychometrics, reliability, validity

## Abstract

The study aimed to assess the reliability of the scores, evidence of validity, and feasibility of the Frail-VIG index. A validation study mixing hospitalized and community-dwelling older people was designed. Intraclass correlation coefficient (ICC) was used to assess the inter-rater agreement and the reliability. The construct validity of the Frail-VIG index with respect to the Frailty Phenotype (FP) was evaluated by calculating the area under the receiver operating characteristic curve (AUC-ROC). Convergent validity with the Clinical Frailty Scale (CFS) was assessed using Pearson’s correlation coefficients. The feasibility was evaluated by calculating the average time required to administer the Frail-VIG index and the percentage of unanswered responses. A sample of 527 older people (mean age of 81.61, 56.2% female) was included. The inter-rater agreement and test–retest reliability were very strong: 0.941 (95% CI, 0.890 to 0.969) and 0.976 (95% CI, 0.958 to 0.986), respectively. Results indicated adequate convergent validity of the Frail-VIG index with respect to the FP, AUC-ROC 0.704 (95% CI, 0.622 to 0.786), and a moderate to strong positive correlation between the Frail-VIG index and CFS (*r* = 0.635, 95% CI, 0.54 to 0.71). The Frail-VIG index administration required an average of 5.01 min, with only 0.34% of unanswered responses. The Frail-VIG index is a reliable, feasible, and valid instrument to assess the degree of frailty in hospitalized and community-dwelling older people.

## 1. Introduction

### 1.1. Background

Over the last few decades, developed countries have undergone a demographic and epidemiological shift that has led to progressive aging of the population and to an increased prevalence of people with chronic diseases [1,2]. While the two most prevalent chronic health problems are multimorbidity and frailty [3], frailty is the chronic condition most frequently associated with poor health outcomes, such as mortality or disability [4,5], as has become apparent during the COVID-19 pandemic [6,7]. In this scenario, the concept of frailty—understood as a vulnerability state against stressing factors due to limited compensatory mechanisms [8]—seems to emerge as a sound line of argument for health systems and their professionals, which require understandable narratives and pragmatic instruments [9,10]. However, despite the widespread consensus regarding the usefulness of the concept of frailty [11] and the need for its routine assessment in the clinical practice [12], there is still some controversy over the operational approach to address it [8].

These difficulties may be explained by two facts. On the one hand, the broadness of the concept of frailty (which ranges from the syndromic view to the accumulation of deficits approach) [13], in addition to the enriching academic debate, may have determined difficulties in its applicability to the healthcare practice. In summary, it can be said that frailty may be presented as a syndromic/dichotomous reality (“*Is this person frail or not?*”) [14], which becomes especially useful for screening for the population that can potentially benefit from preventive actions; the Frailty Phenotype (FP) [14] criteria, the Fatigue, Resistance, Ambulation, Illnesses, and Loss of Weight (FRAIL) [15] questionnaire, the Gérontopôle Frailty Screening Tool [16], or functional performance tests (such as the gait speed test [17] or the Short Physical Performance Battery [18]) are examples of useful instruments for this approach. However, frailty can also be seen as a continuous reality based on the accumulation of different deficits (“*How frail is this person?*”) [19], which is particularly useful to assess a person’s situational diagnosis or degree of reserve [20]. Both the Clinical Frailty Scale (CFS) [21] and the frailty indices (FIs) [22] may be effective instruments in this approach to frailty.

On the other hand, there are many frailty assessment tools available [12], which are not always sufficiently pragmatic or feasible in the daily clinical practice, or which are not valid or reliable enough [23]. In this sense, the psychometric assessment of frailty instruments should be a research priority, in order to produce even stronger evidence on the practical usefulness of the concept of frailty [24,25]. This need becomes especially relevant in the case of FIs [26], for which there are limited studies on reliability of its scores, construct validity, and feasibility [27].

One of the FIs that has shown better mortality predictive capacity is the Frail-VIG index, with an area under the receiver operating characteristic curve (AUC-ROC) of 0.90 and 0.85 at 1 and 2 years, respectively [28,29]. Published in 2017 by Amblàs-Novellas et al., this FI, based on the Comprehensive Geriatric Assessment, consists of 22 trigger questions that are used to assess 25 deficits from eight different dimensions, with a final score that can range from 0 to 1 (with the submaximal limit in the clinical practice being close to 0.7). There is an excel calculator available at https://en.c3rg.com/index-fragil-vig (31 March 2021).

### 1.2. Objective/Rationale

Although previous papers have shown an excellent mortality predictive capacity, as well as good content validity and interpretability, there are no conclusive data on its reliability, construct validity, and feasibility. Therefore, this article aims to analyze the reliability of the scores, evidence of validity, and feasibility of the Frail-VIG index.

## 2. Methods

This article follows the guidelines established by the Consensus-Based Standards for the Selection of Health Measurement Instruments (COSMIN) on the design of studies to assess the measurement properties of instruments [30]. The study protocol was approved by the Ethics Committee of the University Hospital of Vic (2018958/PR189).

### 2.1. Study Design and Participants

This is an observational study, based on the classical test theory [31] and conducted in the prospective FIS/VIG cohort designed for the validation of the Frail-VIG index and the dynamic assessment of frailty over time. Participant recruitment was performed at an intermediate care hospital, with a home-based follow-up of 12 months and quarterly assessments of the degree of frailty by means of the Frail-VIG index.

The inclusion criteria for the study were individuals ≥75 years of age and/or identified as people with complex care needs (PCC, in Catalan) or with palliative care needs (MACA, in Catalan), based on the criteria developed by the Health Department of Catalonia [32,33], who were admitted to the Santa Creu de Vic University Hospital (Barcelona, Spain) during the study enrolment period (July 2018–July 2019). This intermediate care hospital was equipped with 100 beds, as well as subacute care, functional rehabilitation, palliative care, and psychogeriatric units. Patients were admitted from primary care or acute care hospitals, generally in the context of an acute intercurrent process. Those individuals for whom the in-person home follow-up was deemed difficult due to geographical reasons (more than 30 km away from the hospital) were excluded from the study.

### 2.2. Variables and Data Sources

In terms of epidemiological variables, these included age, gender, and usual place of residence. At the clinical level, all the variables included in the Frail-VIG index (Table 1) were collected, as well as the degree of frailty according to the classification into four categories commonly used in our clinical practice: non-frailty (Frail-VIG index score < 0.2), mild frailty (Frail-VIG index score 0.2–0.35), moderate frailty (Frail-VIG index score 0.36–0.5), and severe frailty (Frail-VIG index score > 0.5).

The collection of data at the time of hospitalization was conducted by the hospital’s healthcare professionals (physicians and nurses), with the Frail-VIG index being an instrument used in the regular clinical practice at the Geriatrics and Palliative Care units. Home follow-up upon discharge was performed by four research nurses combining face-to-face visits (months 1, 6, and 12) and telephone visits (months 3 and 9).

### 2.3. Psychometric Assessment of the Frail-VIG Index

This study evaluated the following psychometric parameters: reliability of the scores, evidence of construct validity, and feasibility. The evaluation was performed at different time points (Figure 1).

#### 2.3.1. Reliability

Reliability is the extent to which scores for people who have not changed are the same for repeated measurement under several conditions [34]. Following COSMIN recommendations, the following measures were assessed: (A) inter-rater reliability, by different persons on the same occasion, evaluated by administering the Frail-VIG index with respect to the individual’s baseline situation by two different teams: the geriatrics professionals (physicians and/or nurses) responsible for admission, and by the team responsible for hospitalization of that individual, which was performed blindly (without having the result of the test performed by the other team); (B) test–retest reliability, over time: in this case, the four nurses administered the Frail-VIG index on two separate occasions for about a week in a blind manner (without having the results of the previous test), ensuring similar conditions to the baseline measurement (assessing, in particular, the absence of any added concurrent processes). For the assessment of frailty, the calculation of the internal consistency of the Frail-VIG index was dismissed upon considering it not relevant, given that it was developed as a formative model (in which the items together form the construct) and not as a reflective model (in which all items are a manifestation of the same underlying construct) [35].

#### 2.3.2. Validity

Since previous studies published on the Frail-VIG index have already demonstrated evidence of its content validity and its criterion validity related to mortality [28,29], as well as its convergent discriminative validity related to the EQ-5D-3L index, this study focused on evidence of construct validity between the Frail-VIG index and other frailty measurement tools. To this end, the Frail-VIG index was administered at the same time and in the same subjects at the cohort’s 6 month follow-up, together with the following tools:
As a categorical instrument for the assessment of frailty (frail vs. not frail), the five original FP criteria based on the physical characteristics as reported in the original Cardiovascular Health Study by Fried were used: weight loss, exhaustion, low energy, expenditure, slow walking speed, and weak grip strength [36]. The JAMAR PLUS+ Hand Dynamometer was used to assess grip strength, assessing the average score of two grips of the grip strength of the dominant hand. Those with no characteristics were identified as fit, those with one or two characteristics were identified as pre-frail, and those with three to five characteristics were identified as frail.CFS [21], a validated ordinal measure of frailty based on nine category clinical descriptors and pictographs ranging from one (fit) to nine (terminally ill), was used as a tool to assess continuous frailty.

Although frailty indices assess frailty as a continuous variable, different cutoffs have been proposed in the literature to distinguish between non-frail and frail individuals (≥0.2 [37] vs. ≥0.25 [12]); in some cases, a distinction has also been proposed for non-frail individuals (≤0.08), pre-frail individuals (0.09–0.24), and frail individuals (≥0.25), even weighing the FI result according to the individual’s chronological age [38].

#### 2.3.3. Feasibility

Feasibility measures whether a questionnaire is affordable for use in the environment in which it is intended to be used, and it should be a usual feature in frailty measurements, while also being simple to apply [39]. The two most frequently used measurements are the calculation of percentage of unanswered responses and the time required to administer the measure. To assess the percentage of unanswered responses, the total number of tests performed since the start of the study to the 12 month follow-up was analyzed. To assess the time of administration of the Frail-VIG, the duration of the 12 month home follow-up was timed. Other aspects to consider when assessing feasibility based on COSMIN recommendations that have been incorporated into this study are the education or training required to administer each test, the need for special equipment/devices, and the physical space required [40].

### 2.4. Statistical Methods

Categorical variables are described as frequencies. Quantitative variables are shown as the mean and standard deviation (SD) when the distribution was normal, and as medians with 25th and 75th percentiles when the distribution was asymmetric. We considered a *p*-value <0.05 as statistically significant. The data were analyzed using the latest available version of the IBM SPSS Statistics 27 software.

#### 2.4.1. Reliability

Reliability was assessed using the intraclass correlation coefficients (ICCs) (two-way random) for the inter-rater agreement and test–retest reliability, as well as Bland–Altman plots for their graphical representation. We calculated a minimum requirement of 40 subjects [41], who were randomly selected. ICCs greater than or equal to 0.70 were interpreted as optimal [34].

#### 2.4.2. Validity

In accordance with COSMIN recommendations, we used the AUC-ROC as the method of choice for the assessment of the convergent validity of the Frail-VIG index (continuous score) with respect to the FP (noncontinuous score). AUC-ROCs of <0.70, 0.70–0.89, and ≥0.90 were considered poor, adequate, and excellent, respectively [42]. While there is no gold-standard tool for the assessment of frailty [34], most frailty tools have ended up conducting comparative studies with FP, since it was the first published tool and represented a benchmark for the other initiatives. Thus, for the calibration of the Frail-VIG index with respect to FP, the prevalence of frail individuals was assessed using both instruments, as well as the sensitivity, specificity, positive and negative predictive value, and Youden index for different cutoffs for the identification of a condition of frailty (≥0.20, ≥0.23 and ≥0.25). We also analyzed the discriminative validity of the Frail-VIG index by comparing it between people classified as frail and non-frail using the FP. We hypothesized that people classified as frail would have a substantially higher average index than non-frail people.

On the other hand, the convergent validity between the two continuous score instruments (CFS and Frail-VIG index) was evaluated using Pearson’s correlation coefficients. We expected moderate to strong positive correlations (*r* ≥ 0.50) between the measurement instruments.

#### 2.4.3. Feasibility

Feasibility was evaluated by calculating the average and SD of time required to administer Frail-VIG, as well as the percentage of unanswered responses. To evaluate the time of administration of the Frail-VIG index, a minimum requirement of 40 subjects [41] was estimated, which were randomly selected.

## 3. Results

### 3.1. General Characteristics

A total of 527 individuals were enrolled: 296 (56.2%) women and 231 (43.8%) men, with a mean (SD) age of 81.6 (9.9) years. Table 1 shows the demographic and clinical characteristics of the cohort at the time of enrolment in the study (corresponding to the baseline Frail-VIG index, administered by the team responsible for hospitalization of the subjects), and at the 6 and 12 month follow-ups (administered by the nurses conducting follow-up).

### 3.2. Psychometric Results of the Frail-VIG Index

#### 3.2.1. Reliability

The inter-rater reliability by the two professionals corresponding to the baseline Frail-VIG of the 41 individuals assessed was ICC 0.941 (95% IC, 0.890 to 0.969)—Figure 2A. The test–retest reliability for the 51 individuals assessed was ICC 0.976 (95% CI, 0.958 to 0.986)—Figure 2B. Both results suggest excellent reliability.

#### 3.2.2. Validity

All of the 6 month follow-up subjects were included (*n* = 200). Losses to follow-up with respect to the initial cohort (*n* = 527) corresponded to (1) deaths (227), of which 136 died during hospitalization, mainly (65.4%) in the palliative care unit, (2) definitive losses to follow-up (*n* = 39), (3) and occasional losses to follow-up (n = 62), who were later followed up at 9 months.

Table 2 shows the prevalence of frail individuals in the 6 month follow-up cohort using the FP and the Frail-VIG index for the different cutoffs proposed in the literature (between 0.2 [37] and 0.25 [12]).

When assessing the construct validity of the Frail-VIG index with respect to FP, the AUC-ROC was 0.704 (95% CI, 0.622 to 0.786) (Figure 3), consistently with an adequate convergent validity. Table 3 shows the sensitivity, specificity, positive and negative predictive value, and Youden index for different cutoffs. The Youden index presented its best score (0.43) for the cutoff of the Frail-VIG index at a ≥0.20.

The correlation coefficient between the two continuous score instruments (Frail-VIG index and CFS) for the calculation of their convergent validity showed moderate to strong positive correlation between the Frail-VIG index and CFS (*r* = 0.64, 95% CI, 0.54 to 0.71) (Figure 4).

#### 3.2.3. Feasibility

Of the 2273 tests performed during the first year of follow-up (equivalent to 50,006 variables; 22 variables for each test), the number of missing variables was 170. This is equivalent to 0.34%. Appendix A includes the number of losses of variables in the Frail-VIG index administered in a hospital setting (baseline situation, admission, and discharge) and in the follow-up at community level (1, 3, 6, 9, and 12 months). Losses in this follow-up period at 6 and 12 months correspond to deaths (*n* = 55) or losses to follow-up (*n* = 19). The administration time of 68 individuals was evaluated, with an average of 5.01 min (SD 2.86).

With respect to the more qualitative aspects, a two-session training was conducted for the interviewers, who also had an instruction manual available. For the administration of the Frail-VIG index, no special equipment or physical space was required. In the context of this study, a dynamometer (for grip strength assessment) was only required for the evaluation of convergent validity, as well as a 4 m space and a chronometer to calculate gait speed.

## 4. Discussion

The results obtained support the Frail-VIG index as a reliable, feasible, and valid tool to assess the degree of frailty in hospitalized and community-dwelling older people.

### 4.1. Psychometric Assessment of the Frail-VIG Index

There are limited high-quality reliability, validity, and feasibility data for many of the FI tools. A recent systematic review of the psychometric characteristics of multicomponent tools designed to assess frailty in older adults found that, for example, there were reliability and validity data available for only 21% of the tools [27]. This could be explained by the fact that, as opposed to the Frail-VIG index, many of the frailty assessment tools were developed and tested retrospectively using data available from large-scale longitudinal studies or were developed in conjunction with a larger trial whose main aim was not the development of a frailty assessment tool [27].

#### 4.1.1. Reliability

The reliability of the Frail-VIG index scores can be classified as very strong. There are virtually no previous studies on the reliability of FI [43], which makes it difficult to compare the results obtained. With respect to other frailty instruments, the Frail-VIG index showed better inter-rater reliability (0.94) than, for example, the Edmonton Frail Scale (0.77) [44] or the CFS (from 0.97 [21] to 0.68 [45]).

Test–retest was also excellent (0.97), indicating that if frail elderly people are stable and the Frail-VIG index is administered under similar conditions, their scores remain stable over time. These results are as good as or better than those published for other assessment instruments, such as the CFS (0.87), the Tilburg Frailty Indicator (0.79) [46], or the FRAIL questionnaire (0.71) [47].

#### 4.1.2. Validity

Convergent validity between Frail-VIG index and FP was substantial (0.70), similar to the results previously published by other FIs (0.65) [26]. When the dichotomized Frail-VIG index for the different cutoffs proposed by the literature was compared with the FP, better results of the Youden index were obtained for values ≥0.20, which would endorse it as a cutoff for considering someone frail when using the Frail-VIG index. For this cutoff, the Frail-VIG index showed an overall higher sensitivity (78.3%) than other FIs (45.9 to 60.7%), but a lower specificity (65.0% vs. 83.5 to 90.0%) [26]. However, this assessment of the FIs from a dichotomic perspective is likely to have academic importance rather than clinical implications; clearly, the two measures (FI vs. FP) cannot be considered equivalent, since they are different instruments with different objectives, and the combined/sequential use of both instruments is advisable, as they provide different and complementary clinical information on the individual’s condition [13,48].

Lastly, the degree of correlation between the two instruments that assess frailty as a continuous reality, Frail-VIG index and CFS (*r* = 0.66), was similar to previous studies for these two instruments (*r* = 0.71) [49]. This is consistent with published evidence on convergent validity between CFS and other FIs, ranging from 0.59 of the electronic frailty index [50] to 0.91 of the FI used by Chong et al. [51].

#### 4.1.3. Feasibility

In terms of feasibility, most published studies used administration time as the most common measure, ranging from 44 s for the CFS to 5 to 20 min for the Fried Phenotype [4,52]. In the case of the FI, in the FI-CGA [53], the administration time ranges from 10 [54] or 12.5 min [55] to 25 min [56]. The CSHA-FI [21] requires about 20–30 min [12]. In fact, the time required for the administration of the FIs has been mentioned by some authors as one of the main limitations to their implementation in routine clinical practice [57]. Thus, the Frail-VIG index would fall in the low range of time of administration of the frailty indices, probably as a result of the lower number of variables involved (22), compared, for example, with the more than 30 items of the different versions of the CSHA-FI [21]. There are not many studies either on the completion rate of FI forms. In the study conducted by Lin et al. [55], a completion rate of 45% was found for the FI-CGA, with the majority (91%) of the incomplete forms having minimal amount of data missing—fewer than four items. In this sense, the low number of missing data in our study is remarkable. Lastly, the Frail-VIG index does not require any additional equipment or space, which has sometimes proven an obstacle to the use of CP, GSFT, and some other versions of the FI [55].

### 4.2. Limitations of the Study

The main limitations of the study are probably related to the generalizability of its results; on the one hand, the inclusion criteria determined a relatively large sample of individuals, with a significant degree of frailty. On the other hand, even though this is an instrument designed to be used by both physicians and nurses in all settings, in this study, it was used by geriatrics specialists, who are very familiar with the use of the Frail-VIG index (commonly used in clinical practice in our environment). Thus, for instance, in the assessment of feasibility, both the time of administration of the Frail-VIG index and the low number of missing data could also be explained by the expertise of the professionals involved (there is probably a learning curve in its use by professionals), as well as the thoroughness inherent to the context of a research study.

Another limitation to bear in mind is that not all psychometric properties were assessed in all settings and by all professional profiles. Therefore, for example, reliability and inter-rater reliability were assessed in hospital settings by physicians and nurses, while the test–retest has been performed in a community setting by the follow-up nurses. Thus, more studies are needed to evaluate the psychometric properties in daily clinical practice by other professionals in different settings and populations.

### 4.3. Healthcare Implications and Future Research

It is essential to have reliable, valid, and feasible tools to take advantage of the multiple opportunities offered by the assessment of frailty as a central element of clinical practice, research, and planning in the care of the elderly [9,12,58], ranging from the prevention of disability to the care of individuals with complex and palliative care needs [20,39]. Unfortunately, only 5% of the frailty assessment tools have shown evidence of reliability and validity that was within statistically significant parameters and of fair methodological quality [27].

There are two areas of special interest for future research, which are related to the multidimensional nature of the IF and the assessment of the dynamic behavior of frailty. In the first place, the validity of the content of the instruments should be enhanced with respect to physical, cognitive-psychological, and social frailty [59,60]. In the second place, the serial administration of the Frail-VIG index in a prospective cohort is likely to provide knowledge on the different courses of frailty [61], as well as on the ability of this FI to assess sensitivity to change and responsiveness, understood as the ability of an instrument to distinguish clinically important changes as the result of an intervention [34]. Further studies are also needed to continue to advance in the validation process of the Frail-VIG index, especially with respect to its cross-cultural validity and generalizability.

## 5. Conclusions

According to the COSMIN guidelines, the results obtained endorse the Frail-VIG index as a reliable, feasible, and valid instrument. Firstly, a very strong reliability of the scores was found in its administration among different professionals (inter-rater reliability), as well as in test–retest reliability. Secondly, the Frail-VIG showed a moderate to strong positive correlation with CFS, as well as adequate convergent validity with respect to the FP. This also allowed calibrating the Frail-VIG index for the identification of frail individuals, establishing a frailty threshold at a score of ≥0.20. Lastly, excellent feasibility has been observed in relation to the time of administration, with respect to the few items missed, and due to the lack of specific space or equipment requirements.

All these characteristics, together with their good correlation, with the mortality demonstrated in previous studies, and with the discriminating capacity between the different degrees of frailty, make the Frail-VIG index a particularly interesting tool to assess frail elderly people in hospitalized and community-dwelling settings.

## Figures and Tables

**Figure 1 ijerph-18-05187-f001:**
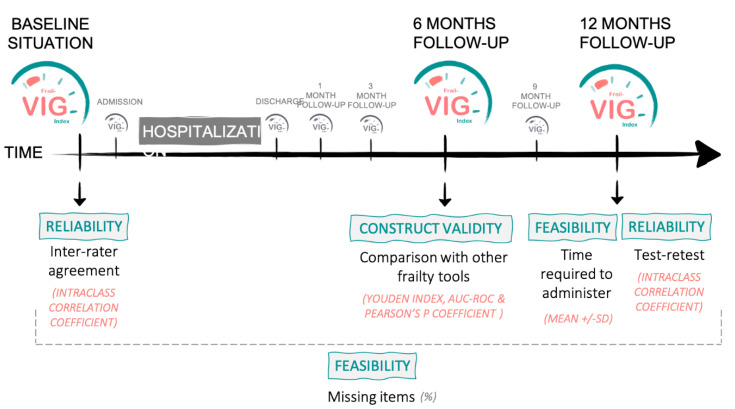
Cohort follow-up timeline, showing the psychometric characteristics assessed at the different moments of the follow-up (as well as the statistical methodology used to assess it).

**Figure 2 ijerph-18-05187-f002:**
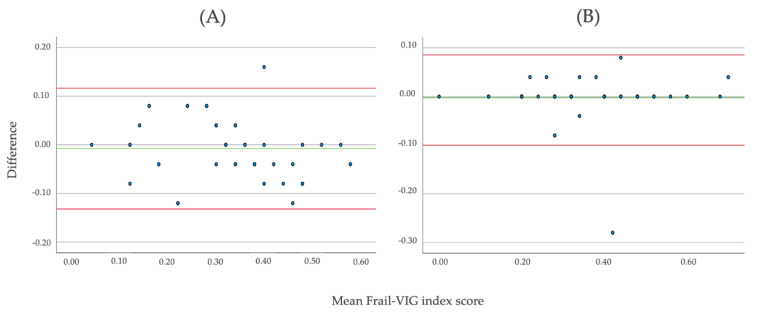
Bland–Altman correlation for the inter-rater reliability (**A**) and test–retest reliability (**B**).

**Figure 3 ijerph-18-05187-f003:**
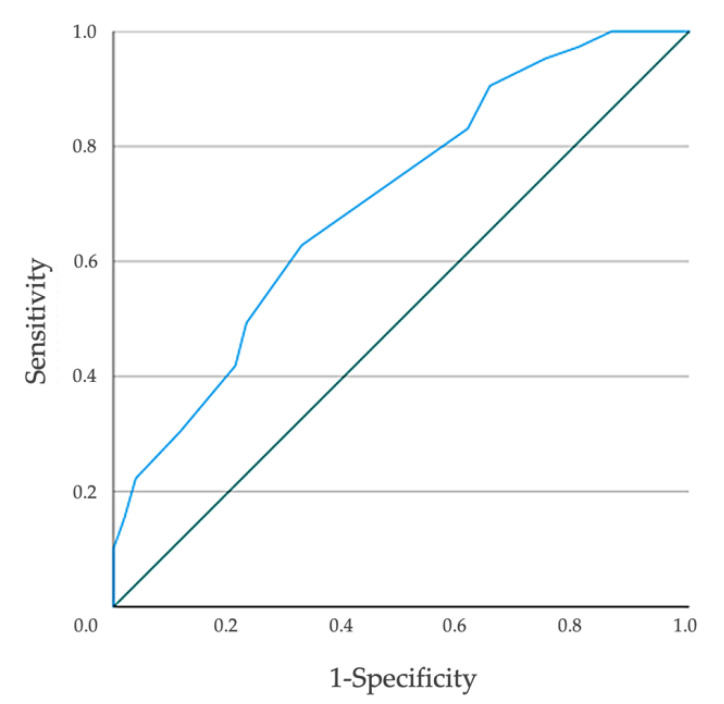
Graphical representation of the ROC plot of the Frail-VIG index, for the people identified as frail according to the Frailty Phenotype criteria.

**Figure 4 ijerph-18-05187-f004:**
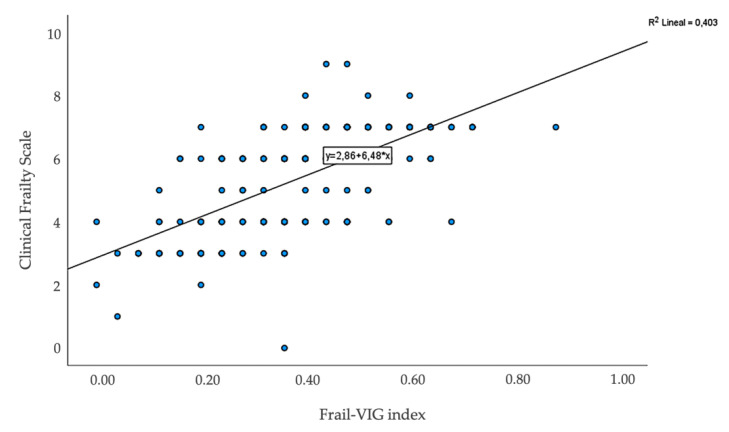
Scatter plots of the correlation between the Frail-VIG index and Clinical Frailty Scale.

**Table 1 ijerph-18-05187-t001:** Epidemiological and clinical characteristics of the cohort at baseline and at the 6 and 12 month follow-ups. At the 6 month follow-up (when the different frailty measurement instruments were compared), the characteristics of the group of non-frail (Frail-VIG index 0–0.19) vs. frail individuals (Frail-VIG index 0.20–1.00) are also shown.

	Baseline Total *N* = 527	Month 6 Follow-Up			Month 12 Follow-Up *n* = 176
		Total *n* = 200	No Frailty *n* = 20 (10.0%)	Frailty *n* = 180 (90.0%)	
**Demographic characteristics**					
Age (years), mean ± SD	81.61 ± 9.9	80.9 ± 10.6	82.6 ± 7.2	80.7 ± 10.9	81.7 ± 9.6
Sex (women), N (%)	296 (56.2)	114 (57.0)	11 (55.0)	103 (57.2)	98 (55.7)
**Usual habitat, *No* (%)**					
Nursing home	68 (12.9)	63 (31.5)	0 (0.0)	63 (35.0)	48 (27.3)
Home	440 (83.5)	129 (64.5)	20 (100.0)	109 (60.5)	111 (63.1)
Others	2 (0.4)	1 (0.5)	0 (0.0)	1 (0.6)	0 (0.0)
Missing information	17 (3.2)	7 (3.5)	0 (0.0)	7 (3.9)	17 (9.6)
**Living arrangement ^1^, *No* (%)**					
With family	303 (68.5)	93 (71.6)	15 (75.0)	78 (70.9)	76 (68.5)
With caregiver	22 (5.0)	6 (4.6)	0 (0.0)	6 (5.5)	5 (4.5)
Alone	105 (23.8)	25 (19.2)	4 (20.0)	21 (19.1)	18 (16.2)
Others	4 (0.9)	0 (0.0)	0 (0.0)	0 (0.0)	0 (0.0)
Missing information	8 (1.8)	6 (4.6)	1 (5.0)	5 (4.5)	12 (10.8)
**Frail-VIG variables**					
Functional IADLs (0–3), mean ± SD	1.48 ± 1.3	1.81 ± 1.2	0.15 ± 0.3	1.99 ± 1.1	1.80 ± 1.3
Barthel index (0–100), mean ± SD	73.87 ± 27.5	57.5 ± 32.4	90.5 ± 22.5	53.9 ± 31.3	62.2 ± 31.1
Malnutrition, N (%)	144 (27.3)	34 (17.1)	2 (10.0)	32 (17.9)	15 (9.1)
Cognitive impairment, N (%)	198 (37.6)	83 (41.7)	0 (0.0)	83 (46.3)	62 (37.8)
Depressive syndrome, N (%)	165 (31.3)	78 (39.6)	2 (10.0)	76 (42.9)	75 (44.9)
Insomnia/anxiety, N (%)	255 (48.4)	119 (59.8)	4 (20.0)	115 (64.2)	96 (56.8)
Social vulnerability, N (%)	74 (14.0)	4 (2.0)	0 (0.0)	4 (2.0)	8 (4.8)
Delirium, N (%)	85 (16.1)	59 (29.5)	1 (5.0)	58 (32.2)	45 (26.8)
Falls, N (%)	111 (21.1)	35 (17.7)	1 (5.0)	34 (19.1)	23 (13.9)
Ulcers, N (%)	56 (10.6)	27 (13.5)	0 (0.0)	27 (15.0)	17 (10.4)
Polypharmacy, N (%)	425 (80.6)	176 (88.0)	357 (86.7)	176 (88.0)	141 (83.9)
Dysphagia, N (%)	88 (16.7)	41 (20.6)	0 (0.0)	41 (22.9)	28 (17.2)
Pain, N (%)	131 (24.9)	62 (31.0)	2 (10.0)	60 (33.3)	36 (21.7)
Dyspnoea, N (%)	47 (8.9)	34 (17.1)	1 (5.0)	33 (18.4)	21 (12.8)
Cancer, N (%)	128 (24.3)	43 (21.8)	2 (10.0)	41 (23.2)	25 (15.3)
Chronic respiratory disease, N (%)	147 (27.9)	78 (39.4)	3 (15.0)	75 (42.1)	63 (37.3)
Chronic cardiac disease, N (%)	232 (44.1)	111 (55.5)	6 (30.0)	105 (58.3)	89 (53.9)
Chronic neurological disease, N (%)	74 (14.1)	47 (23.5)	2 (10.0)	45 (25.0)	34 (20.5)
Chronic digestive disease, N (%)	40 (7.6)	39 (20.1)	2 (10.0)	37 (21.2)	28 (17.2)
Chronic renal disease, N (%)	210 (39.8)	91 (46.4)	3 (15.8)	88 (49.8)	68 (40.7)
**Frailty degree ^2^**					
Total cohort average, mean ± SD	0.31 ± 0.15	0.39 ± 0.16	0.11 ± 0.05	0.42 ± 0.14	0.35 ± 0.16
No frailty, N (%)	115 (21.8)	20 (10.0)	20 (100.0)	-	35 (20.5)
Mild frailty, N (%)	190 (36.1)	52 (26.0)	-	52 (28.9)	43 (25.1)
Intermediate frailty, N (%)	147 (27.9)	77 (38.5)	-	77 (42.8)	59 (34.5)
Severe frailty, N (%)	75 (14.2)	51 (25.5)	-	51 (28.3)	34 (19.9)

^1^ Refers to patients not living in a nursing home. ^2^ The frailty degree was calculated using the categorization of the Frail-VIG index into no frailty (Frail-VIG index score <0.2), mild frailty (Frail-VIG index score 0.2–0.35), moderate frailty (Frail-VIG index score 0.36–0.5), and advanced frailty (Frail-VIG index score >0.5). IADLs, Instrumental Activities of Daily Living; SD, standard deviation.

**Table 2 ijerph-18-05187-t002:** Prevalence of frail people in the cohort using Frailty Phenotype (FP), as well as different cutoffs of the Frail-VIG index.

		Non-Frailty/Pre-Frailty	Frailty
FP	No (%)	52 (26.0)	148 (74.0)
	Frail-VIG, mean ± SD	0.30 (0.16)	0.42 (0.15)
IF-VIG	No (%)	20 (10)	180 (90)
(Frailty cutoff ≥0.20)	Frail-VIG, mean ± SD	0.11 (0.05)	0.42 (0.14)
IF-VIG	No (%)	32 (16)	168 (84)
(Frailty cutoff ≥0.23)	Frail-VIG, mean ± SD	0.14 (0.06)	0.44 (0.13)
IF-VIG	mean ± SD	45 (22.5)	155 (77.5)
(Frailty cutoff ≥0.25)	Frail-VIG, mean (DS)	0.17 (0.07)	0.45 (0.12)

**Table 3 ijerph-18-05187-t003:** Sensitivity, specificity, and positive and negative predictive value between the Frail-VIG index and the Frailty Phenotype (FP).

	Sensitivity	Specificity	PPV	NPV	Youden Index	Frail-VIG Index (Frailty Value Cutoff)
**FP**	78.3%	65.0%	95.3%	25.0%	0.43	≥0.20
	79.8%	56.3%	90.5%	34.6%	0.36	≥0.23
	79.4%	44.4%	83.1%	38.5%	0.24	≥0.25

NPV, negative predictive value; PPV, positive predictive value.

## Data Availability

The data presented in this study are available on request from the corresponding author.

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
