# Peer review of "Reliability, Validity, and Feasibility of the Frail-VIG Index"

_ijerph, 2021, doi:10.3390/ijerph18105187_

Round 1

Reviewer 1 Report

The Originality of the manuscript is relevant. Despite being an extension of the analysis of the validity of a tool for the diagnosis of frailty, it had to be contemplated and comes to fill that space. However, there are some aspects that must be clarified or at least justified:

It should be explained how the recruitment and follow-up of the participants has been carried out, since, although the inclusion criteria are indicated, the high number of participants for a 100-bed hospital (initial n 527), and, above all, the need Home monitoring performed from the hospital should be clarified. Likewise, in this section it must be specified whether the described cohort is representative of the general population.

I believe that it should be justified why the type of coexistence of the participants has not been considered, since according to numerous studies this has a high relationship with frailty and personal autonomy.

On the other hand, it is necessary to know whether the consequences on the IF-VIG of the intercurrent process that led to the income - which probably could have led to an increase in the value of the IF-VIG - have been considered.

The results are adequate and show coherence with the objective of the study. The scientific soundness is enough and the analyzes carried out allow us to achieve the objectives.

Only additional clarifications would be appreciated, so that other researchers can try to reproduce the results:

- Clarify the average time it takes to pass the questionnaire analyzed.
- Such a high fragility result among the study population is surprising. Clarify whether this high rate may have led to an overstatement of the discrimination capacity of the test.

Regarding the quality of the presentation, the article is adequately written, but there is a formal aspect that must necessarily be corrected: the numbering of the different sections of the text generates confusion when repeating the numbers of the epigraphs and their title (for example 2.3.2 and 2.4.2 have the same name; or section 2.3.2 appears as Validity and also at the same time as Feasibility).

Reviewer 2 Report

References: 14,18,19,21,35,41 are over 15 years ago , but after the analysis, I think they can stay at work.

I believe this paper to fulfil requirements of the periodical and I recommend it for publication.

Reviewer 3 Report

The authors present a validation study of a frailty index assessment tool (Frail-VIG Index). The instrument appears simple to administer, has a calculator that provides the frailty index, and has been evaluated in a large population of individuals. The article is very well written, the statistical analyses are rigorous and the discussion and conclusions adequately address the results found. The instrument is very interesting for assessing the degree of severity, allows monitoring progression over time and can help in clinical decision making.

Introduction

Please provide a clearer definition of frailty.

Objective

Properly employ psychometric terms, reliability is of the scores not the test, evidence of validity, rather than validity. You may refer to:

http://www.psicothema.com/psicothema.asp?id=4508

The objectives are too general, you should specify them in concrete objectives and order them according to their presentation in the results section.

Method

It would be clearer if, in the participants section, the number of participants and some basic sociodemographic data (mean age, sex) were described.

Report the country where the study was conducted.

Replace "validity" with "evidence of validity".

Results

The results are described with high precision

Discussion

You do not need to report the statistical data again in this section, simply assessing the differences between your results and those found by other authors is sufficient.

Minors

Specify acronyms the first time they are mentioned. E.g. FRAIL, CFS...
